# LLM Unlearning Reveals a Stronger-Than-Expected Coreset Effect in Current Benchmarks

**Soumyadeep Pal**[†,⋆]  **Changsheng Wang**[†,⋆]  **James Diffenderfer**[‡]
**Bhavya Kailkhura**[‡]  **Sijia Liu**[†,§]
[†]Michigan State University, [‡]Lawrence Livermore National Laboratory, [§]IBM Research
[⋆]Equal contribution

## Abstract

Large language model (LLM) unlearning has become a critical challenge in ensuring safety and controlled model behavior by removing *undesired* data-model influences from the pretrained model while preserving its general utility. Significant recent efforts have been dedicated to developing LLM unlearning benchmarks such as WMDP (Weapons of Mass Destruction Proxy) and MUSE (Machine Unlearning Six-way Evaluation), facilitating standardized unlearning performance assessment and method comparison. Despite their usefulness, we uncover for the first time a novel *coreset effect* within these benchmarks. Specifically, we find that LLM unlearning achieved with the original (full) forget set can be effectively maintained using a significantly smaller subset (functioning as a "coreset"), *e.g.*, as little as 5% of the forget set, even when selected at random. This suggests that LLM unlearning in these benchmarks can be performed surprisingly easily, even in an extremely low-data regime. We demonstrate that this coreset effect remains strong, regardless of the LLM unlearning method used, such as NPO (Negative Preference Optimization) and RMU (Representation Misdirection Unlearning), the popular ones in these benchmarks. The surprisingly strong coreset effect is also robust across various data selection methods, ranging from random selection to more sophisticated heuristic approaches. We explain the coreset effect in LLM unlearning through a keyword-based perspective, showing that keywords extracted from the forget set alone contribute significantly to unlearning effectiveness and indicating that current unlearning is driven by a compact set of high-impact tokens rather than the entire dataset. We further justify the faithfulness of coreset-unlearned models along additional dimensions, such as mode connectivity and robustness to jailbreaking attacks. Codes are available at https://github.com/OPTML-Group/MU-Coreset.

## 1 Introduction

The problem of machine unlearning (MU) for large language models (LLMs), referred to as LLM unlearning (Liu et al., 2025; Si et al., 2023; Qu et al., 2024; Cooper et al., 2024), is gaining critical importance as a means of enforcing data privacy rights (*e.g.*, preventing the generation of copyrighted or sensitive content) (Eldan & Russinovich, 2023; Shi et al., 2024b; Maini et al., 2024; Jang et al., 2022), and for removing harmful or unsafe knowledge embedded in models amid growing concerns around safety and alignment (Li et al., 2024; Yao et al., 2024; Barez et al., 2025; Zhang et al., 2024b; Chen et al., 2025). The core objective of MU is to remove the influence of specific, undesired data or knowledge from a trained model, while preserving its general utility, without the cost of full retraining from scratch.

Despite the growing importance of LLM unlearning, much of the existing research has primarily focused on the design of unlearning *algorithms*. Notable approaches include gradient ascent and its variants (Thudi et al., 2022; Jang et al., 2022; Yao et al., 2024), which aim to reverse the training effect of the forget data by explicitly pushing the model away from learned patterns; influence function-based methods (Izzo et al., 2021; Jia et al., 2024b; Koh & Liang, 2017), which assess and mitigate the contribution of individual data points to

model behavior; preference optimization techniques and their extensions (Rafailov et al., 2023; Zhang et al., 2024a; Fan et al., 2024b), which guide the model to reduce its preference for responses associated with the forget data; misrepresentation learning methods (Li et al., 2024), which disrupt internal representations linked to the undesired knowledge; and neuron- or model-editing approaches that leverage task vectors or localized model components to guide unlearning (Jia et al., 2024a; Hase et al., 2023; Wu et al., 2023).

While the above LLM unlearning algorithms have advanced the field, the *data perspective* of LLM unlearning, such as how the composition or size of the forget set (*i.e.*, the dataset used to define the unlearning scope) influences unlearning performance, has received significantly less attention. On the data side, some recent efforts explored evaluating the memorization level of forget data to better guide unlearning optimization (Barbulescu & Triantafillou, 2024; Zhao et al., 2024). Yet, most existing efforts focused on developing LLM unlearning *benchmark datasets*, such as TOFU (Task of Fictitious Unlearning) to remove synthesized fictitious information (Maini et al., 2024), WMDP (Weapons of Mass Destruction Proxy) for harmful knowledge removal (Li et al., 2024) and MUSE (Machine Unlearning Six-way Evaluation) for copyrighted data removal (Shi et al., 2024b) and other similar datasets (Eldan & Russinovich, 2024; Liu et al., 2024).

Although the above benchmarks have served as standardized platforms for LLM unlearning training and evaluation, the forget datasets curated in these benchmarks are typically provided as fixed, full sets, and existing research has largely adopted them without questioning the appropriateness, sufficiency, or redundancy of these forget sets for achieving effective unlearning. For example, in WMDP, a large forget data source is provided containing sensitive documents (25K articles) related to hazardous knowledge in biosecurity and cybersecurity. However, the forget set actually used in the WMDP-based unlearning via RMU (Representation Misdirection Unlearning) utilizes only the first 600 articles from these source documents (Li et al., 2024), leaving open the question of whether the full dataset is necessary or if a much smaller subset could suffice. Therefore, in this work, we ask:

> *(Q) Does there exist a "coreset" within the LLM unlearning forget dataset that can yield lossless unlearning performance?*

The term "coreset" is inspired by prior research on coreset selection for non-LLMs (Guo et al., 2022; Borsos et al., 2020; Zhang et al., 2023), where a subset of the training set enables comparable model performance to training on the full dataset. We envision that addressing (Q) could open up a new, underexplored paradigm: LLM unlearning in the low-data regime, where effective unlearning is achieved using only a small subset of the forget data. It may also reveal potential weaknesses in current unlearning benchmarks (Thaker et al., 2024a), where unlearning could be surprisingly easy due to redundancy in the forget set.

In this work, we provide an affirmative answer to (Q) through extensive empirical studies: a coreset effect does exist in LLM unlearning, and unlearning in current benchmarks such as WMDP and MUSE can be effectively achieved using as little as 5% of the original full forget set, even when the coreset is randomly selected. We summarize **our contributions** below.

• We unveil a novel *Coreset Effect* in LLM unlearning, showing that popular unlearning methods (RMU and negative preference optimization) achieve comparable performance on benchmarks like WMDP and MUSE using as little as 5% of the forget set.

• We demonstrate that this effect holds across both random and more sophisticated coreset selection strategies. We also provide a keyword-based analysis to explain the strong coreset effect where small forget subsets can drive unlearning behavior.

• We show that coreset-unlearned models exhibit similar unlearning characteristics to full-set-unlearned ones across dimensions including mode connectivity, adversarial robustness, and model utility, except for a potential increased vulnerability to relearning attacks.

## 2   Related Works

**LLM Unlearning.** Unlearning seeks to remove the influence of undesired data-model influence to protect privacy or prevent harmful knowledge generation. A commonly-used gold standard is exact unlearning–retraining the model from scratch without the forget

data (named "Retrain") (Cao & Yang, 2015; Thudi et al., 2022) –but this is computationally prohibitive, especially for LLMs. As a result, significant efforts have focused on *approximate unlearning* (Bourtoule et al., 2021; Liu et al., 2025), which modifies pretrained model weights post hoc. These approaches fall broadly into model finetuning methods (Ilharco et al., 2022; Li et al., 2024; Zhang et al., 2024a; Fan et al., 2024b; Jia et al., 2024a), and input-based prompting methods (Pawelczyk et al., 2023; Thaker et al., 2024b). Recent studies have also highlighted vulnerabilities in these methods, including jailbreaking (Łucki et al., 2024; Lynch et al., 2024), latent extraction (Seyitoğlu et al., 2024), and fine-tuning-based relearning attacks (Hu et al., 2024; Deeb & Roger, 2024), revealing fragility in unlearning robustness. Despite this progress, Thaker et al. (2024a) argues that current benchmarks offer an *overly optimistic view* due to weak evaluation protocols. As a complementary concern, we uncover a strong coreset effect in standard LLM unlearning benchmarks, showing that effective unlearning can be achieved using surprisingly small subsets of the forget data, which raises potential questions about the reliability and rigor of current unlearning dataset designs.

**Data perspective in unlearning.** Recent work has begun to explore the influence of forget data on unlearning outcomes, particularly through the lens of data ordering and importance. For instance, Zhao et al. (2024) introduced RUM, a meta-unlearning algorithm that partitions the forget set into homogeneous subsets using data quality scores, such as representation distance from the centroid and memorization score, and applies specialized unlearning strategies to each subset sequentially. Zhao & Triantafillou (2024) extended this framework by evaluating additional memorization proxies for sample partitioning. Both of these studies are limited to image classification tasks. In the context of LLMs, Barbulescu & Triantafillou (2024) proposed a dynamic unlearning approach that iteratively selects high-memorization samples from the forget set in each epoch. While these efforts highlight the role of data quality or ordering in unlearning, they do not examine the existence of a coreset.

**Coreset selection.** Coreset selection, developed mainly for classification using computer vision models, refers to the process of selecting a subset of the training data, which can maximize the model accuracy on said task when trained using such subset. One class of coreset selection methods typically assign importance scores to each sample in the dataset based on various metrics computed during training dynamics such as gradient norm (Paul et al., 2021), l2 norm of error vectors (Paul et al., 2021), forgetting score (Toneva et al., 2019), data loss (Welling, 2009; Pruthi et al., 2020) and prediction confidence (Pleiss et al., 2020). Additionally some methods also focus on data diversity, based on distances of datapoints from their class clusters (Xia et al., 2022) and coverage of the training set (Zheng et al., 2022; Maharana et al., 2024). Another class of methods are optimization based approaches for coreset selection mainly focusing on gradient information of data samples (Mirzasoleiman et al., 2020; Killamsetty et al., 2021a;b; Pooladzandi et al., 2022).

For language models, alongside loss-based scoring (Azeemi et al., 2023), various classic scoring-based coreset selection methods (mentioned above) have been explored for downstream finetuning tasks where a smaller model is used for efficient scoring (Zhang et al., 2025). Additionally, coreset selection methods have been developed for instruction-tuning where the influence of each training sample is computed using gradient similarity with samples in the validation set (Xia et al., 2024) and using LLM-as-a-Judge (Chen et al., 2024; Liu et al., 2023). Finally, Zhou et al. (2023) demonstrated superior performance LLM-alignment by finetuning on an extremely small set of manually curated high quality dataset.

To the best of our knowledge, Patil et al. (2025) concurrently explores coreset selection in LLM unlearning. Using anomaly detection on the hidden state representations of the forget set, the authors prune about 10-30% of the forget set to determine the coreset with the primary goal of utility preservation. However, our work differs from this work in two key aspects: (1) Their study did not identify the strong coreset effect embedded in existing LLM unlearning benchmarks, nor did it characterize its conditions (*e.g.*, the relationship between coreset effect and unlearning training time); And (2) we go further by investigating both the "what" and "why" behind the coreset effect, focusing on extreme pruning ratios (90-99%), *i.e.*, extreme low-data regimes (1–10%), while preserving unlearning and utility performance, and analyzing its impact on unlearning robustness.

## 3 Preliminaries and Motivation on Coreset Effect in LLM Unlearning

**LLM unlearning setup.** LLM unlearning aims to remove undesired data, knowledge, or model influence from a pretrained LLM, while preserving its general utility. This is typically framed as a model fine-tuning or editing task, given access to a forget set $\mathcal{D}_f$, which contains the data or knowledge to be unlearned, and a retain set $\mathcal{D}_r$, which is typically unrelated to the unlearning target and serves to help preserve the model's overall utility post-unlearning. Therefore, the problem of LLM unlearning is cast as the following regularized optimization framework (Liu et al., 2025; Yao et al., 2024; Maini et al., 2024),

$$\min_{\boldsymbol{\theta}} \ \mathbb{E}_{(x,y)\in\mathcal{D}_f}[\ell_f(y|x;\boldsymbol{\theta})] + \lambda\mathbb{E}_{(x,y)\in\mathcal{D}_r}[\ell_r(y|x;\boldsymbol{\theta})], \tag{1}$$

where $\boldsymbol{\theta}$ denotes the model parameters to be updated during unlearning (initialized from the pretrained model state), $\mathcal{L}_f$ and $\mathcal{L}_r$ denote the forget loss (*i.e.*, the unlearning objective) and retain loss (*i.e.*, the utility-preserving objective), respectively, evaluated when using $\boldsymbol{\theta}$ to generate a response $y$ given input $x$, and $\lambda > 0$ is a regularization parameter to strike the balance between the forget and retain objectives.

**Representative LLM unlearning methods considered in this study.** The specifics of the unlearning optimization in (1) correspond to different LLM unlearning approaches, typically depending on how the forget loss $\ell_f$ is formulated and implemented. Although numerous unlearning methods have been proposed in the literature, our work focuses on two widely used and high-performing LLM unlearning approaches, negative preference optimization (**NPO**) (Zhang et al., 2024a) and representation misdirection unlearning (**RMU**) (Li et al., 2024), where NPO typically excels at data-wise unlearning, as demonstrated on the MUSE benchmark (Shi et al., 2024b), and RMU is particularly effective for knowledge unlearning, as evidenced by its strong performance on the WMDP benchmark (Li et al., 2024). NPO is adapted from direct preference optimization (DPO) (Rafailov et al., 2023), but differs by treating the forget data in $\mathcal{D}_f$ as negative examples, excluding any positive examples. This yields a preference-based forget loss $\ell_f$ that drives the unlearned model $\boldsymbol{\theta}$ to deviate from the original, pre-trained model (referred to as the reference model in NPO) on the forget set $\mathcal{D}_f$. The retain loss $\ell_r$ in NPO is typically chosen as the standard cross-entropy next-token prediction loss on the retain set $\mathcal{D}_r$. In contrast to NPO, RMU is built upon a random feature-based unlearning objective (Li et al., 2024; Fan et al., 2024a). The forget loss $\ell_f$ in RMU promotes unlearning by mapping the intermediate-layer representations of the forget data to random features, thereby disrupting the model's ability to generate or reconstruct the forgotten information. In a similar spirit, the retain loss $\ell_r$ enforces alignment between the representations of the unlearned model and the pre-unlearned (reference) model on the retain dataset, helping to preserve the model's general utility. We refer readers to Appendix A for the detailed formulations of NPO and RMU.

**Unlearning benchmarks considered in this study.** We conduct and evaluate LLM unlearning using two benchmarks: **WMDP** (Li et al., 2024), which assesses the unlearning of potentially hazardous knowledge in domains such as biology (called **WMDP-Bio**) and cybersecurity (called **WMDP-Cyber**); and **MUSE** (Shi et al., 2024b), which focuses on unlearning textual content from Harry Potter books (named **MUSE-Books**) and news articles (named **MUSE-News**). We select these two benchmarks as the former is the representative *knowledge* unlearning benchmark and the latter is designed to additionally support *data* unlearning (*e.g.*, to prevent verbatim memorization and privacy leakage). It is worth noting that other unlearning benchmarks also exist, such as TOFU (Maini et al., 2024). However, the evaluation of unlearning effectiveness in TOFU relies on a *p*-value metric, which can be insensitive in distinguishing unlearning performance when the *p*-value exceeds 0.1, making it less effective for fine-grained comparisons between unlearning methods.

In WMDP (Li et al., 2024), the goal of unlearning LLM is to reduce its *accuracy on the WMDP evaluation set*, which comprises QA pairs with undesirable answers related to harmful knowledge targeted for removal. Accordingly, we measure **unlearning effectiveness** (**UE**) as: UE = 100% − (Accuracy on WMDP evaluation set). Since the associated QA pairs are multiple-choice questions with 4 choices, a perfect unlearning process would achieve an UE of 75%. The **utility** (**UT**) of the WMDP-unlearned model is measured by the zero-shot accuracy on the *MMLU* dataset (Hendrycks et al., 2020).

In MUSE (Shi et al., 2024b), **UE** is measured using different metrics: (1) *Verbatim memorization* (**VerbMem**) on the forget set $\mathcal{D}_f$ reflects the model's ability to perform next-token prediction for completing the forgotten data records. (2) *Knowledge memorization* (**KnowMem**) reflects the model's ability to answer questions involving undesired knowledge in MUSE. Thus, a lower VerbMem (or KnowMem) indicates better UE, as it implies reduced model generation capability for the targeted data (or knowledge) removal. Besides VerbMem and KnowMem, UE in MUSE is also evaluated using (3) *privacy leakage* (**PrivLeak**), which assesses the extent to which the unlearned model leaks membership information, *i.e.*, whether it reveals that data in $\mathcal{D}_f$ was part of the original training set. PrivLeak values approaching zero indicate better unlearning. **UT** of the unlearned model is measured by **KnowMem on MUSE's retain set** $\mathcal{D}_r$, reflecting the model's ability to preserve useful knowledge unrelated to unlearning.

Furthermore, we note that when implementing NPO and RMU to solve problem (1) for WMDP and MUSE, the number of training epochs is typically kept *small*. This is because the forget loss encourages divergence from the pretrained model, and prolonged training may lead to model collapse or degradation of general utility. For example, *the number of unlearning epochs for RMU is set to 1* for WMDP and MUSE-Books. We refer the readers to Appendix B for a detailed discussion of the unlearning settings across different benchmarks.

**Problem of interest: "Coreset" effect in LLM unlearning under low-data regimes.** In current unlearning setups, a key pre-condition is the assumption of access to the forget set $\mathcal{D}_f$, which is typically provided *a priori* in current benchmarks. However, very few studies have investigated the problem of LLM unlearning in a low-data regime, *i.e.*, when the size of the forget set $\mathcal{D}_f$ is limited or small. Therefore, the primary problem of interest in this work is to explore whether a "coreset"-like effect exists in LLM unlearning, thereby enabling effective unlearning in low-data regimes of current benchmarks.

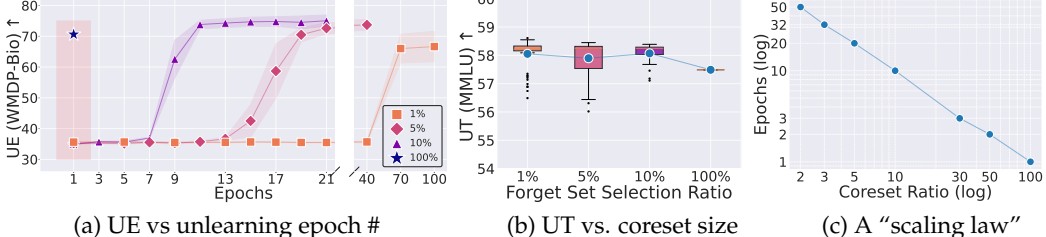

(a) UE vs unlearning epoch #          (b) UT vs. coreset size          (c) A "scaling law"

Figure 1: Unveiling the "coreset"-like effect in LLM unlearning on WMDP using the RMU method, applied to the pre-trained LLM Zephyr-7B-$\beta$. The "coreset" is randomly sampled from the full forget set ($\mathcal{D}_f$ = WMDP-Bio), with selection ratios of 1%, 5%, and 10%; the 100% setting corresponds to using the entire $\mathcal{D}_f$. Unlearning performance is averaged over 5 random trials. (a) The "coreset" achieves comparable UE (unlearning effectiveness) for RMU on WMDP-Bio, especially when unlearning is performed with longer training epochs. The default number of unlearning epochs is 1 for RMU under the full $\mathcal{D}_f$ (100%), as indicated by the shaded region. (b) MMLU-based UT (utility) of the unlearned model against the forget set selection ratio. Each box plot represents the UT performance of the unlearned model across the range of unlearning epochs shown in (a) for 5 random trials. (c) A linear "scaling law" in the log-domain between the unlearning epochs and the forget set selection ratio for lossless unlearning.

Through the motivating results in **Fig. 1**, we demonstrate the existence of a coreset effect– where a small random subset of the original forget set has been able to achieve comparable test-time unlearning performance, *especially when unlearning is trained for extended epochs*. To be specific, Fig. 1 presents the unlearning performance in terms of both UE (unlearning effectiveness) and UT (utility measured on MMLU), using the RMU unlearning method applied to the pre-trained LLM Zephyr-7B-$\beta$ on the WMDP benchmark. The performance is evaluated across different random data selection ratios (1%, 5%, and 10%) with respect to the full forget set WMDP-Bio. Fig. 1(a) shows that an unlearned model trained on a small subset of the forget set (*e.g.*, as little as 5% of the full set) can achieve UE almost the same as that of using the entire forget set (100%). However, **this coreset effect only emerges when the training is extended beyond the RMU's default 1-epoch setting on WMDP**; For example, using only 1% of the forget set requires over 70 training epochs to achieve comparable UE. In addition, the observed effectiveness of coreset unlearning does **not** come at the cost of UT. This is evidenced by Fig. 1(b), where the unlearning optimization (via

RMU) maintains UT on MMLU across varying unlearning epochs (indicated by the box regions) and coreset ratios, in comparison to the model unlearned using the 100% forget set. In this case of lossless unlearning with a coreset (*i.e.* with UE and UT comparable to the full forget set), we demonstrate an unlearning scaling law in Fig. 1(c). This shows how we need to gradually increase the number of unlearning epochs as the coreset ratio decreases.

The above motivating results provide strong evidence for a coreset effect in LLM unlearning, showing that with sufficient training, small randomly selected subsets can achieve comparable UE and UT to full forget set-based unlearning. This finding prompts several key questions: *Does the effect hold or even improve with non-random coreset selection? What underlying factors drive this phenomenon?* And *are coreset-unlearned models as faithful and robust as their full-set counterparts?* We investigate these questions in the following sections.

## 4 On the Consistency and Rationale Behind Forget Coreset Effect

In this section, we present a comprehensive study of the coreset effect in LLM unlearning across various dimensions, including different unlearning approaches (RMU and NPO), benchmarks (WMDP and MUSE), and coreset selection methods. Furthermore, we explore a possible explanation for this effect through the lens of keywords extracted from the coreset, which may serve as a key driving force behind effective unlearning.

**Surprisingly strong and consistent performance of *random* coreset selection.** Recall from our motivating results on WMDP-Bio unlearning using RMU in Fig. 1, that coresets formed via random data selection already demonstrate lossless UE while maintaining UT. To further validate the effectiveness of random selection (termed RANDOM), we investigate another two key aspects: (1) random coreset unlearning across different unlearning methods and benchmarks, and (2) comparison with other more sophisticated coreset selection methods.

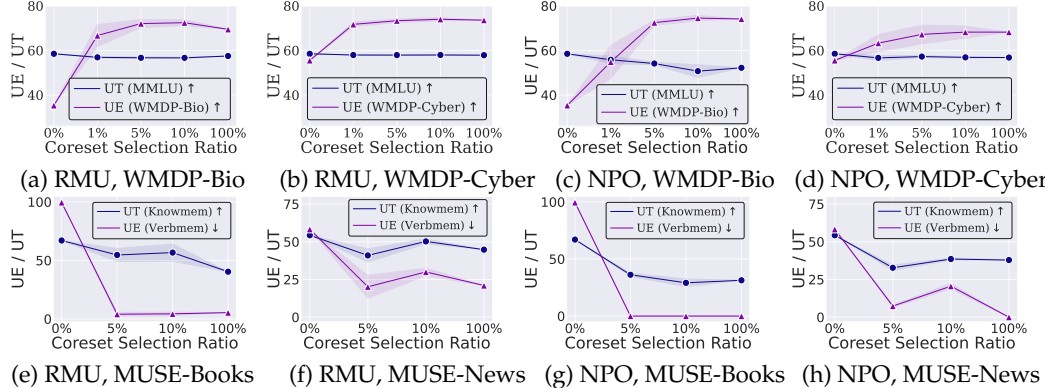

(a) RMU, WMDP-Bio  (b) RMU, WMDP-Cyber  (c) NPO, WMDP-Bio  (d) NPO, WMDP-Cyber

(e) RMU, MUSE-Books  (f) RMU, MUSE-News  (g) NPO, MUSE-Books  (h) NPO, MUSE-News

Figure 2: Consistent RANDOM-based coreset unlearning performance in terms of UT and UE across against the coreset selection ratio. The performance is averaged over 5 independent trials for random coreset selection, with variance indicated by the shaded regions. (a)-(d) correspond to the results of applying a specific unlearning method (RMU or NPO) to a benchmark dataset (WMDP-Bio, WMDP-Cyber, MUSE-Books, or MUSE-News). Following the benchmark setting, unlearning is performed using Zephyr-7B-$\beta$ on WMDP, LLaMA2-7B on MUSE-News, and ICLM-7B on MUSE-Books.

**Fig. 2** shows the consistent RANDOM-enabled coreset effect in LLM unlearning using RMU and NPO across four datasets: WMDP-Bio, WMDP-Cyber, MUSE-Books, and MUSE-News. The coreset selection ratio ranges from 0% to 100%, where 0% corresponds to the original (pre-unlearning) model, and 100% represents standard unlearning using the entire forget set. In Fig. 2(a)–(d), which shows WMDP-based unlearning, we observe that both RMU and NPO exhibit a clear coreset effect, evidenced by lossless UE and preserved UT as the coreset ratio increases comparable to that achieved with the full forget set (100%). Notably, RMU appears more effective in the low-data regime, achieving strong performance with as little as 1% of the forget set, whereas NPO requires around 5% to reach similar results in WMDP. As shown for MUSE-based unlearning in Fig. 2(e)–(h), both RMU and NPO exhibit a consistent coreset effect in the context of data unlearning, as reflected by reductions in verbatim memorization. Notably, UT, measured by KnowMem, even improves when using a smaller coreset size; see Fig. 2(e).

Next, we compare the unlearning performance of RANDOM-based coreset selection with three additional, more sophisticated coreset selection methods, GRAND (Paul et al., 2021), MODERATE (Xia et al., 2022), and MIN-K% PROB (Shi et al., 2024a). The first two methods are classic coreset selection techniques adapted for LLMs, where GRAND ranks forget samples based on the gradient norm of the unlearning objective in (1), and MODERATE clusters forget samples based on their associated deep representations, then ranks them by their distance to the cluster center. MIN-K% PROB was originally developed to determine whether a given text appears in the original pretraining dataset, producing a data memorization score, where a higher value indicates stronger memorization of the data. In our setting, we select the unlearning coreset based on the top memorization scores. We refer readers to Appendix C for details in the above coreset selection methods.

Table 1: Coreset unlearning performance (UE and UT, consistent with Fig. 2) using RMU and NPO on WMDP-Bio and WMDP-Cyber, evaluated using Zephyr-7B-$\beta$ across varying coreset selection ratios (0%, 5%, 10%, 100%) and selection methods (RANDOM, GRAND, MODERATE, MIN-K% PROB). Here 0% and 100% refer to the pre-unlearning case (w/o using any forget data) and the standard unlearning case (w/ the full forget set), respectively. The performance for RANDOM-based selection is reported in the form $a \pm b$, where $a$ is the mean and $b$ is the standard deviation, computed over 5 independent trials. The performance of other data selection metrics are reported using the mean over 2 trials.

| Coreset Ratio | Unlearning Method | RMU on WMDP-Bio | | NPO on WMDP-Bio | | RMU on WMDP-Cyber | | NPO on WMDP-Cyber | |
|---|---|---|---|---|---|---|---|---|---|
| | | UE (↑) | UT (MMLU) (↑) | UE (↑) | UT (MMLU) (↑) | UE (↑) | UT (MMLU) (↑) | UE (↑) | UT (MMLU) (↑) |
| 0% | No unlearning | 35.35 | 58.48 | 35.35 | 58.48 | 55.51 | 58.48 | 55.51 | 58.48 |
| 100% | Full forget set | 69.46 | 57.48 | 74.11 | 52.20 | 73.57 | 57.85 | 68.19 | 56.79 |
| 10% | RANDOM | $72.43_{\pm1.34}$ | $56.66_{\pm0.24}$ | $74.50_{\pm1.22}$ | $50.69_{\pm2.85}$ | $73.96_{\pm0.74}$ | $57.95_{\pm0.15}$ | $68.21_{\pm2.69}$ | $56.88_{\pm0.35}$ |
| | GRAND | 71.30 | 56.98 | 71.54 | 52.40 | 73.43 | 57.25 | 67.64 | 56.71 |
| | MODERATE | 70.86 | 56.66 | 75.92 | 52.36 | 73.53 | 57.20 | 69.65 | 56.69 |
| | MIN-K% PROB | 70.61 | 56.86 | 74.59 | 52.71 | 73.91 | 57.12 | 69.19 | 57.03 |
| 5% | RANDOM | $72.03_{\pm1.78}$ | $56.69_{\pm0.44}$ | $72.47_{\pm1.23}$ | $54.12_{\pm0.56}$ | $73.32_{\pm0.79}$ | $57.89_{\pm0.12}$ | $67.19_{\pm4.17}$ | $57.23_{\pm0.43}$ |
| | GRAND | 72.59 | 56.78 | 75.19 | 56.55 | 72.88 | 57.64 | 65.85 | 56.58 |
| | MODERATE | 73.78 | 56.90 | 71.72 | 56.18 | 73.56 | 57.52 | 67.47 | 55.72 |
| | MIN-K% PROB | 69.48 | 57.29 | 70.91 | 55.67 | 72.84 | 57.58 | 69.15 | 57.19 |

**Table 1** presents the coreset unlearning performance across different coreset selection methods on WMDP. As we can see, the coreset effect remains robust across different coreset selection methods. Although random selection introduces performance variance across trials due to differing coreset realizations, more sophisticated selection strategies typically offer only marginal improvements in UE or UT over RANDOM, and these gains generally fall within the variance observed under RANDOM. Notably, compared to standard unlearning (*i.e.*, using the full 100% forget set), 5% coreset unlearning can consistently achieve lossless UE and UT, demonstrating its effectiveness even in low-data regimes. This consistent performance is also observed on the MUSE benchmark, as shown in **Table D1**.

**Explaining the sufficiency of coreset unlearning: A keyword perspective.** To explain the consistently strong coreset-like effect, we analyze the unlearning behavior of RANDOM-based coreset selection from a keyword perspective. Given a coreset, we extract its corresponding keyword set by leveraging LLM-as-a-judge (OpenAI o1 (Jaech et al., 2024)), guided by our proposed prompt (detailed in Appendix E), to identify words most relevant to the targeted unlearning concept or knowledge (*e.g.*, biosecurity). We then perform unlearning on this keyword set to assess whether the majority of the unlearning efficacy can be attributed to this significantly smaller subset, in comparison to unlearning conducted on the full coreset (from which the keywords were extracted) and the original full forget set $\mathcal{D}_f$.

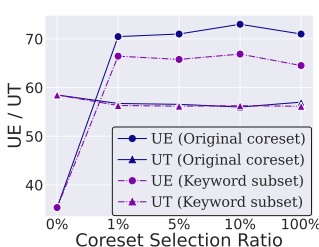

Figure 3: Unlearning performance (UE and UT) using the original coreset and its keyword subset across varying coreset selection ratios for RMU-based unlearning on (WMDP-Bio, Zephyr-7B-$\beta$).

**Table 2** presents representative keyword examples and compares the generation of unlearned models across three different forget sets: the full $\mathcal{D}_f$, a 5% coreset, and a keyword-only subset extracted from the coreset. The keyword set is constructed by filtering out non-keyword tokens and preserving the original order, as highlighted in the first column. As we can see, unlearning with just the keyword set remains highly effective, comparable to

Table 2: Generation of different unlearned models for the sentence completion task using a part of the forget set of WMDP-Bio as input. Keywords are marked in light cyan within the forget sample. The unlearning task and setting follow the same configuration as in Fig. 2(a).

| Forget data sample from $\mathcal{D}_f$ (WMDP-Bio) w/ extracted keywords | Full Forget Set ($\mathcal{D}_f$) Unlearned Model Generation | Coreset (5% of $\mathcal{D}_f$) Unlearned Model Generation | Keywords Unlearned Model Generation |
|---|---|---|---|
| The most common pathogen isolated from urine cultures is Escherichia coli , 80–90%. However, other bacteria that were rarely isolated previously are now rising ( Proteus , Citrobacter , Enterobacter , and Serratia species). E.coli can produce extended-spectrum β-lactamase ( ESBL ) enzymes , which provide resistance against drugs like penicillins , extended-spectrum cephalosporins , and monobactams . These ESBL-producing bacteria are associated with __ | \n \nMany, there),"""v,\n""B,s.\n Where they\nshould \n@(R) Rory()\n \nffa\nwhere \nJ} .\nB \n(L.)\nT\n \nR)Mart\n\n###more\n\n for\n n#(v)\n\n\n\nthere\n\nNone \n\n\nto\n \nto\n\n to \n\nwhere\n\nto \n \nhad\n\nto\n\nbased\n\n | –could to réve γ révère, ρ β B, τ α χρε ντεσβα, ... | 'sometimes' were 'few' in-levels.\n\n'The' is a 'zero'-levels. \n\n'The' is 'zero'- 'μ'-'n'-'i'-'o'-'t' -'100'-'n'-'x'-'i'- ... |
| Since their first use as expression vectors in the 1980s, Ad vectors have received tremendous attention as gene delivery vehicles for vaccine antigens . They have been extensively tested as vaccine delivery systems in several pre- clinical and clinical studies for a number of infectious diseases including measles , hepatitis-B , rabies , anthrax , Ebola , severe acute respiratory syndrome ( SARS ), human immunodeficiency virus 1 ( HIV-1 ), malaria , tuberculosis , and influenza . There are two basic types of Ad vectors that are being utilized for gene delivery applications. The first type of Ad vectors __ | .@the...\n \n###.more\n\n.\n \n###.s\n vin.x \n.\n\n##,\ninway.\nx.\n \n@\n@@ in.\n ###\nxer\nmore\n\nb\n no\n\nw \n\nThere\n@x\nh\n\n no\n\n scundo\nthere.\n\nmo\n\n \n how.\n \n Cural.\nsair\n\nhg\n .\n' | .\n\n(none)\n\n.there.. iron. törko.you.unde.undef. und. undund@.sUnd.none\n \nundo .@.every.all.they.just\n \nwWx.wereunder.undo.yer .@.\al.@x.@\n \nnrex.undund. universo@.under.all \n.there.under.rem@.unpret' | who knew, were, and, and, were, and, and, were, were, and, were, were, were, and, were, were, and, were, were, were, were, were, were, ... |

the full and coreset-based unlearning, as the model fails to recall the target knowledge and instead generates nonsensical responses to forget queries.

For a thorough quantitative evaluation, **Fig. 3** compares the unlearning performance (in both UE and UT) of using the keyword-only coreset and the original coreset against the coreset selection ratio on WMDP-Bio, following the same unlearning task setting as in Fig. 2(a). We perform 300 epochs of unlearning for the keyword-only forget sets. As observed, although unlearning with the keyword-only forget sets does not fully match the UE of the full coreset, it captures the majority of the unlearning effect compared to the pre-unlearning baseline at 0% coreset selection ratio. The contribution of using keyword unlearning is consistent against the coreset selection ratio. This suggests that the coreset effect may be driven by a small number of high-impact keywords within the coreset, which are sufficient to account for most of the unlearning performance. This, in turn, points to a surprising simplicity in the current unlearning benchmark tasks or their evaluation protocols. Additional analysis on keyword overlap between the coreset and full forget set is provided in Appendix F.

## 5 On the Faithfulness of LLM Unlearning using Coresets

In this section, we investigate the quality and faithfulness of coreset-based LLM unlearning from additional perspectives: (1) mode connectivity, which reflects the similarity of the unlearning loss landscape to that of full forget set-based unlearning; (2) the robustness of coreset-unlearned models against both input-based jailbreaking attacks and weight-based model fine-tuning post unlearning; and (3) additional utilities of unlearned LLMs beyond those captured by existing unlearning benchmarks. Unless otherwise specified, in this section we focus on LLM unlearning on WMDP using RANDOM-based coreset selection, following the settings used in Sec. 4.

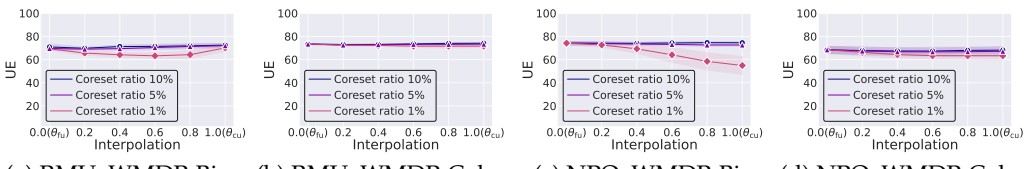

(a) RMU, WMDP-Bio  (b) RMU, WMDP-Cyber  (c) NPO, WMDP-Bio  (d) NPO, WMDP-Cyber

Figure 4: LMC holds between coreset-unlearned model ($\theta_{cu}$) and the full forget set-unlearned model ($\theta_{fu}$), as evidenced by UE against the interpolation coefficient $\alpha$ (x-axis). Here the coreset-unlearned models are obtained using RANDOM-based coresets with the same setting as in Fig. 2(a-d).

**Mode connectivity: Coreset unlearning is as good as full forget set unlearning.** Mode connectivity refers to the phenomenon where the minima found by two optimized ML models are connected by a path along which the model error does not increase, suggesting that the models reside in a shared or smoothly connected region of the loss landscape (Draxler et al., 2018; Freeman & Bruna, 2016; Qin et al., 2022; Garipov et al., 2018). A strong form of connectivity is known as linear mode connectivity (**LMC**), where the interpolation path between two models is constrained to be linear in parameter space (Frankle et al., 2020). Therefore, we leverage LMC to investigate the similarity or potential discrepancy

between a coreset-unlearned model ($\boldsymbol{\theta}_{cu}$) and the full forget set-unlearned model ($\boldsymbol{\theta}_{fu}$). **LMC for coreset unlearning is said to hold** if the unlearning evaluation (*e.g.*, via UE) of the *linearly interpolated model* $\boldsymbol{\theta}(\alpha) := (\alpha\boldsymbol{\theta}_{cu} + (1-\alpha)\boldsymbol{\theta}_{fu})$ remains approximately consistent as the interpolation coefficient $\alpha \in [0, 1]$ varies, where $\boldsymbol{\theta}(1) = \boldsymbol{\theta}_{cu}$ and $\boldsymbol{\theta}(0) = \boldsymbol{\theta}_{fu}$ yield the endpoints of the linear path. **Fig. 4** illustrates UE of the interpolated model $\boldsymbol{\theta}(\alpha)$ against the interpolation coefficient $\alpha$ at forget coreset selection ratios of 1%, 5%, and 10%. As we can see, the UE remains *approximately constant* along the linear interpolation path between $\boldsymbol{\theta}_{cu}$ and $\boldsymbol{\theta}_{fu}$, indicating a *strong* LMC between the two models (nearly perfect connectivity for 5% and 10% coresets). This provides strong evidence that the coreset-unlearned model resides in the same optimal basin as the full forget set-unlearned model, supporting the faithfulness of unlearning achieved via coresets.

**Adversarial robustness against jailbreaking attacks.** The lack of robustness in LLM unlearning has been highlighted by their vulnerability to jailbreaking attacks (Łucki et al., 2024; Lynch et al., 2024; Patil et al., 2024). Therefore, we investigate whether coreset-based unlearning introduces additional robustness limitations compared to unlearning with the full forget set. **Table 3** presents the UE of coreset-unlearned models, alongside the full forget set-unlearned

Table 3: Robustness of models unlearned using RANDOM coreset selection on WMDP-Bio using RMU under Zephyr-7B-$\beta$, following the setting in Fig. 2(a). Robustness is measured using the UE reduction after enhanced GCG attack.

| Coreset Ratio | UE | | UE reduction |
|---|---|---|---|
| | **Before Attack** | **After Attack** | **After Attack** |
| 100% | 69.46 | 47.71 | 21.75 |
| 10 % | $72.43_{\pm1.34}$ | $53.39_{\pm0.02}$ | 19.04 |
| 5 % | $72.03_{\pm1.78}$ | $51.29_{\pm0.03}$ | 20.74 |

model (*i.e.*, coreset ratio 100%), under input-level jailbreaking attacks using the enhanced-GCG method (Łucki et al., 2024), which generates adversarial prompt prefixes to elicit forgotten information. A lower UE after the attack indicates a more successful recovery of forgotten content. As shown, the coreset-unlearned models experience a similar degree of UE reduction as the full-set counterpart, demonstrating comparable robustness. These results further support the faithfulness and reliability of coreset-based unlearning.

**Robustness against downstream fine-tuning.** Beyond jailbreaking attacks, unlearning robustness has also been studied in model fine-tuning, where forgotten knowledge can resurface in unrelated downstream tasks (Hu et al., 2024; Deeb & Roger, 2024; Lo et al., 2024), similar to such vulnerability of safety-aligned LLMs (Qi et al., 2023). We finetune models unlearned on WMDP using NPO using GSM8k (Cobbe et al., 2021) and AG-News (Zhang et al., 2015) using 600 samples from the finetuning dataset. In **Fig. 5**, we demonstrate UE of unlearned models subjected to such fine-tuning. As expected, the models exhibit 'relearning' of unlearned information. However, in certain scenarios, such as WMDP-Bio unlearned

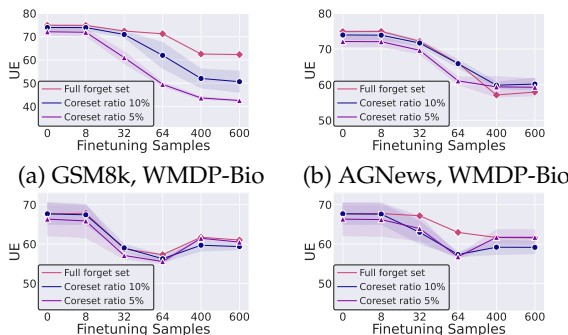

(a) GSM8k, WMDP-Bio    (b) AGNews, WMDP-Bio

(d) GSM8k, WMDP-Cyber (e) AGNews, WMDP-Cyber

Figure 5: Unlearning performance (UE) of RANDOM-coreset unlearned models (using NPO under Zephyr-7B-$\beta$) against the number of fine-tuning samples. (a)-(f) Relearning using finetuning datasets (GSM8k, AGNews) for models unlearned on WMDP-Bio or WMDP-Cyber. The performance is averaged over 3 independent trials.

models finetuned using GSM8k in Fig. 5(a), the speed and extent of relearning can clearly distinguish coreset-unlearned models from those unlearned using the full forget set. These results highlight a potential downside of coreset unlearning: *using a smaller forget set may compromise robustness to downstream fine-tuning compared to full-set unlearning.* This is likely because, when the relearn set is larger than coreset, the influence of relearning can overpower the unlearning.

In Appendix G, we also show that coreset-unlearned models retain utility on auxiliary tasks such as math addition/subtraction and TruthfulQA.

# 6    Conclusion and Discussion

We establish a novel coreset effect in LLM unlearning, where an extremely small subset of the forget set can achieve lossless unlearning. This coreset effect particularly emerges when conditioned on sufficient unlearning training. And this effect is shown to hold consistently across benchmarks (WMDP, MUSE) and unlearning methods (RMU, NPO). We further explain and validate this phenomenon from multiple perspectives, including keyword analysis, mode connectivity, and robustness. While we establish this effect, we observe that many classic heuristic-based coreset selection methods do not outperform simple RANDOM selection. Therefore, determining the 'optimality' of random selection or identifying optimized, improved coresets remains a critical direction for future research. Our keyword analysis also highlights the need for deeper investigation into the underlying mechanisms of unlearning, potentially through more advanced interpretability techniques. In other words, unlearning could be driven by salient tokens within the forget data, rather than relying on the entire forget dataset. Additionally, the strong coreset effect observed in this work suggests potential 'redundancy' within the forget sets used in current benchmarks. We hypothesize that this effect arises from inherent 'correlations' among forget data points. If the forget examples were independently and identically distributed, the coreset effect might diminish significantly. This correlation-driven redundancy could make unlearning tasks *easier than expected* under current evaluation protocols. We advocate for future benchmarks to incorporate the influence of forget dataset size and richer evaluation methods to better reflect realistic unlearning challenges.

## Ethics Statement

The ethical implications of our work on identifying the coreset effect and developing coreset-based unlearning methods lie in promoting responsible data handling and the safe use of large language models (LLMs). Our research advances machine unlearning techniques aimed at strengthening privacy protections, reducing sociotechnical harms, and ensuring more controlled, trustworthy generative AI. At the same time, we recognize that the societal impact of unlearning is nuanced and continually evolving. This underscores the importance of ongoing reflection, transparent evaluation, and active engagement with the broader discourse on AI ethics and governance. On the other hand, it is equally important to prevent the misuse of unlearning techniques to erase safe, beneficial, or ethically important information from LLMs, which could compromise model integrity and safety.

## Acknowledgments

We thank the U.S. Department of Energy via Lawrence Livermore National Laboratory (LLNL) under Contract DE-AC52-07NA27344 and the LLNL LDRD Program under Project No. 23-ER-030 for their support (LLNL-JRNL-2004686). Soumyadeep Pal, Changsheng Wang, and Sijia Liu were also partially supported by the National Science Foundation (NSF) CISE Core Program Award (IIS-2207052), the NSF CAREER Award (IIS-2338068), the ARO Award (W911NF2310343), and the Amazon Research Award for AI in Information Security. We also thank Dr. Alfred O. Hero for his invaluable suggestions, which has helped us improve our work.

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

## A Details of NPO and RMU

In this section, we provide details explaining the two main unlearning methods we consider in this paper.

**Negative Preference Optimization (NPO).** NPO (Zhang et al., 2024a) reduces the model's preference for the forget set $\mathcal{D}_f$ by treating it analogously to negative responses in preference optimization, but omitting the positive response term. This yields the following loss function, which has been shown to achieve lower divergence rates compared to various other unlearning approaches:

$$\ell_{\text{NPO}}(\boldsymbol{\theta}) = \mathbb{E}_{(x,y)\in\mathcal{D}_f} \underbrace{\left[ -\frac{2}{\beta} \log \sigma \left( -\beta \log \left( \frac{\pi_{\boldsymbol{\theta}}(y|x)}{\pi_{\text{ref}}(y|x)} \right) \right) \right]}_{\ell_f \text{ specified in (1)}} \tag{A1}$$

where $\sigma(t) = 1/(1 + e^{-t})$ is the sigmoid function, $\beta > 0$ is a hyperparameter. Minimizing the above forget loss drives the model to be unlearned $\pi_{\boldsymbol{\theta}}$ *away* from the reference model $\pi_{\text{ref}}$ for the samples from the forget set. In NPO, the retain loss $\ell_r$ is simply the prediction loss *i.e.* the cross entropy loss between $x$ and $y$, which are obtained from $\mathcal{D}_r$.

**Representation Misdirection Unlearning (RMU).** RMU (Li et al., 2024) seeks to degrade the model's internal representations for the forget set, thereby reducing its ability to recall or generate undesired knowledge. Specifically, it aligns the hidden state activations of the unlearned model at a given layer $l$ (denoted $\pi_{\boldsymbol{\theta}}^l$) with a random unit vector u, where each component of u is sampled independently and uniformly from $[0, 1)$. In contrast to NPO, RMU also includes a retain loss that ensures the model's representations on the retain dataset remain consistent with those of the reference (pre-unlearned) model. Considering $z_f$ and $z_r$ to be samples from the forget and retain set respectively, this is formalized as follows:

$$\ell_{\text{RMU}}(\boldsymbol{\theta}) = \mathbb{E}_{z_f\in\mathcal{D}_f} \underbrace{\left[ \frac{1}{L_f} \sum_{\text{token } t\in z_f} \|\pi_{\boldsymbol{\theta}}^l(t) - c\cdot\mathbf{u}\|_2^2 \right]}_{\ell_f \text{ specified in (1)}} + \lambda \mathbb{E}_{z_r\in\mathcal{D}_r} \underbrace{\left[ \frac{1}{L_r} \sum_{\text{token } t\in z_r} \|\pi_{\boldsymbol{\theta}}^l(t) - \pi_{\text{ref}}^l(t)\|_2^2 \right]}_{\ell_r \text{ specified in (1)}} \tag{A2}$$

where $L_f$, $L_r$ are the number of tokens in $z_f$, $z_r$ respectively and $c$ controls the strength of **u**.

## B Detailed Unlearning Setup

We present the detailed unlearning setups for different settings in **Table B1**. It is worth noting that the first 600 samples of WMDP-Bio and WMDP-Cyber are used as the *full* forget set following (Li et al., 2024). For NPO, we perform unlearning for 140 steps with a batch size of 4, which is approximately $(140 \times 4)/600 = 0.93$ epochs.

Table B1: Full forget set unlearning settings across different unlearning methods and benchmarks.

| Unlearning Benchmark | Unlearning Method | Model | Epochs | Learning Rate |
|---|---|---|---|---|
| WMDP-Bio/Cyber | RMU NPO | Zephyr-7B-$\beta$ | 1 0.93 | $5e-5$ $7e-6$ |
| MUSE-Books | RMU NPO | ICLM-7B | 1 | $1e-3$ $1e-5$ |
| MUSE-News | RMU NPO | LLaMA2-7B | 10 | $1e-3$ $1e-5$ |

Unless otherwise mentioned, we follow the standard hyperparameters in the above benchmarks. For MUSE-Books (Shi et al., 2024b), the reference model ICLM-7B is finetuned on Harry Potter books, while for MUSE-News, the reference model LLaMA2-7B is finetuned on BBC News articles. These reference models are available in the MUSE benchmark.

As mentioned in Sec. 3, the coreset effect emerges with a higher number of unlearning epochs. In **Table B2**, we present the epoch numbers for each coreset setting. All other unlearning settings are the same as the full forget set unlearning.

Table B2: Coreset unlearning settings across different unlearning methods and benchmarks.

| Unlearning Benchmark | Unlearning Method | Coreset Ratio | Epochs |
|---|---|---|---|
| WMDP-Bio/Cyber | RMU | 10%
5%
1% | 10
20
100 |
| WMDP-Bio/Cyber | NPO | 10%
5%
1% | 9.33
18.67
93.33 |
| MUSE-Books | RMU | 10%
5% | 10
20 |
| MUSE-Books | NPO | 10%
5% | 7
8 |
| MUSE-News | RMU | 10%
5% | 100
200 |
| MUSE-News | NPO | 10%
5% | 5
12 |

## C  Details of Coreset Selection Methods

In this section, we present additional details of the heuristic data selection methods as presented in Sec. 4. Though these methods were primarily developed for non-LLMs, we adapt them to our unlearning process as detailed below.

**GRAND.**   The main rationale behind this method (Paul et al., 2021) is that the importance of each sample $z_f$ from the forget set is captured by the expected gradient norm of the loss associated with that sample, where the expectation is taken over the unlearning trajectory. Thus the GRAND score is given by:

$$\chi(z_f) = \mathbb{E}_{\theta_t} \|\nabla_{\theta_t}[\ell_f(z_f; \theta_t) + \lambda \ell_r(z_r; \theta_t)]\|_2 \; ; \text{ where } z_f \sim \mathcal{D}_f, \; z_r \sim \mathcal{D}_r \tag{C1}$$

Here $\ell_f$ and $\ell_r$ are specified in (1), where $\ell_f$ changes according to the unlearning method as specified before. In our experiments, we consider the trajectory of unlearning for 10 epochs for the purpose of computing the above expectation.

**Moderate.**   The moderate coreset selection method (Xia et al., 2022) was developed in the classification setting where samples were divided into clusters according to their classes. Using a well-trained model, the class center of each class is calculated using the hidden state representations of the penultimate layer of the corresponding samples. Since our setting does not involve class labels, we cluster the forget samples into four groups using K-means, based on their penultimate-layer representations extracted from the reference (pre-unlearned) model. For each cluster, we compute its centroid and rank the samples by their distance to the respective centroid. To select representative data points, we choose those whose distances are closest to the median within their cluster.

**MIN-K% PROB.**   The MIN-K% PROB method (Shi et al., 2024a) was developed to ascertain whether a given text appears in the original pretraining dataset. We use this metric as a data selection method for unlearning under the notion that data points not encountered during training are less influential in the unlearning process. In fact, in (Shi et al., 2024a), the authors demonstrated that the MIN-K% metric can be used as an effective indicator for assessing removal of knowledge after unlearning.

For a given datapoint $z$, we calculate the log-likelihood of each token and choose a set of top 40%(K = 40) of tokens with lowest value, called Min-K%($z$). Then the score for $z$ is calculated as:

$$\text{MIN-K\% PROB}(z) = \frac{1}{|\text{Min-K\%}(z)|} \sum_{z_i \in \text{Min-K\%}(z)} \log p(z_i | z_1, ..., z_{i-1}; \theta) \tag{C2}$$

After computing the above scores for each sample $z$ in $\mathcal{D}_f$, we choose samples with the top p scores to obtain our coresets. Thus, if we want to choose a 5% coreset, then p $= 0.05 \times |\mathcal{D}_f|$.

# D Coreset Unlearning Performance on MUSE

**Table D1** presents the coreset unlearning performance ofor MUSE across different coreset selection methods. Here we observe that performance of RANDOM-based coresets are highly comparable to the full forget set performance, while also strongly outperforming the full forget set in many instances. For example, we observe a significant utility (UT) improvement for MUSE-Books using RMU for coresets as seen in Table D1, col.6. In fact, for MODERATE-based 10% coreset we see ∼ 12% improvement in utility over the full forget set.

Additionally, we observe that for MUSE-News, the heuristic based 5% coreset selection methods have UE (Knowmem) similar to that of the full forget set (as evidenced by Table D1, col.3). However, this is worse than the average performance of RANDOM-based selection. This coupled with the high variance of the performance RANDOM-based coresets points us to the fact that coreset selection is a non-trivial problem for such cases. Nevertheless, the performance of RANDOM-based coreset supports our strong coreset observation even at 5% coreset selection regime.

Table D1: Coreset-based unlearning performance (UE and UT, consistent with Fig. 2) using RMU and NPO on MUSE evaluated using LLaMA2-7B on News and ICLM-7B on Books. The table is presented in the same format as **Table 1**. Here 'Retrain' refers to a model finetuned only on MUSE excluding the forget set.

| Coreset Ratio | Unlearning Method | RMU | | | | NPO | | | |
|---|---|---|---|---|---|---|---|---|---|
| | | UE | | | UT | UE | | | UT |
| | | VerbMem (↓) | KnowMem (↓) | PrivLeak (→ 0) | KnowMem (↑) | VerbMem (↓) | KnowMem (↓) | PrivLeak (→ 0) | KnowMem (↑) |
| **MUSE-Books** | | | | | | | | | |
| 0% | No unlearning | 99.56 | 58.32 | -56.32 | 67.01 | 99.56 | 58.32 | -56.32 | 67.01 |
| 0% | Retrain | 14.30 | 28.90 | 0.00 | 74.50 | 14.30 | 28.90 | 0.00 | 74.50 |
| 100% | Full forget Set | 5.38 | 18.72 | -16.98 | 40.34 | 0.00 | 0.00 | -31.02 | 31.33 |
| 10% | RANDOM | $4.34_{\pm1.40}$ | $23.86_{\pm10.79}$ | $-12.79_{\pm4.86}$ | $56.67_{\pm7.29}$ | $0.00_{\pm0.00}$ | $0.00_{\pm0.00}$ | $-27.09_{\pm2.22}$ | $29.12_{\pm3.36}$ |
| | GRAND | 4.96 | 21.69 | 5.07 | 57.11 | 0.00 | 0.00 | -25.28 | 37.54 |
| | MODERATE | 5.72 | 9.02 | -19.84 | 33.44 | 0.00 | 0.00 | -28.42 | 36.55 |
| | MIN-K% PROB | 7.53 | 20.12 | -29.17 | 56.43 | 0.00 | 0.00 | -0.24 | 33.21 |
| 5% | RANDOM | $4.01_{\pm1.58}$ | $19.37_{\pm6.75}$ | $-18.61_{\pm1.88}$ | $54.62_{\pm5.54}$ | $0.00_{\pm0.00}$ | $3.48_{\pm0.56}$ | $-8.97_{\pm8.03}$ | $36.19_{\pm1.35}$ |
| | GRAND | 5.69 | 28.00 | -37.66 | 59.21 | 0.00 | 3.77 | -15.71 | 40.15 |
| | MODERATE | 4.75 | 7.08 | -18.02 | 33.92 | 0.00 | 3.35 | -5.63 | 36.93 |
| | MIN-K% PROB | 4.20 | 37.48 | -0.12 | 67.76 | 0.00 | 4.33 | -32.90 | 39.18 |
| **MUSE-News** | | | | | | | | | |
| 0% | No unlearning | 58.29 | 62.93 | -98.71 | 54.31 | 58.29 | 62.93 | -98.71 | 54.31 |
| 0% | Retrain | 20.75 | 33.32 | 0.00 | 53.79 | 20.75 | 33.32 | 0.00 | 53.79 |
| 100% | Full forget set | 21.00 | 54.68 | 1.80 | 44.75 | 0.00 | 47.96 | 107.40 | 37.86 |
| 10% | RANDOM | $29.97_{\pm2.50}$ | $18.85_{\pm21.99}$ | $-38.87_{\pm2.78}$ | $50.23_{\pm1.67}$ | $20.54_{\pm1.71}$ | $42.30_{\pm1.26}$ | $-99.58_{\pm0.14}$ | $38.54_{\pm0.61}$ |
| | GRAND | 28.85 | 40.50 | -45.84 | 49.50 | 21.35 | 41.97 | -99.71 | 38.59 |
| | MODERATE | 32.62 | 38.40 | -36.44 | 52.77 | 19.26 | 42.46 | -99.78 | 37.99 |
| | MIN-K% PROB | 31.67 | 2.02 | -15.95 | 50.14 | 23.59 | 43.85 | -98.14 | 40.66 |
| 5% | RANDOM | $20.15_{\pm7.50}$ | $12.95_{\pm17.17}$ | $-16.42_{\pm18.70}$ | $40.93_{\pm4.32}$ | $7.36_{\pm0.79}$ | $34.28_{\pm1.92}$ | $-76.38_{\pm7.46}$ | $32.75_{\pm1.75}$ |
| | GRAND | 24.15 | 50.26 | -33.27 | 47.11 | 3.51 | 32.71 | -1.85 | 31.65 |
| | MODERATE | 19.37 | 52.21 | -8.53 | 47.00 | 3.19 | 38.24 | 16.50 | 35.57 |
| | MIN-K% PROB | 28.83 | 49.21 | -33.74 | 43.94 | 3.40 | 36.45 | 73.07 | 31.21 |

# E   Prompt to Extract Keywords

---

**Prompt to Extract Keywords using LLM-as-a-Judge**

**System**:

You are an expert in chemistry, biology and diseases. I will provide you with a text, and your task is to carefully analyze each word and determine if it is related to chemistry or biology and disease. For every word that is related to chemistry, biology and disease, replace it with an asterisk (*) in the original text while keeping all other words unchanged. Think step by step and ensure accuracy in identifying only chemistry, biology and disease-related terms. Pay close attention to scientific terminology. Return the result in the following format:
{{
"replaced_text": "<text after replacing>",
"related_words": ["word1", "word2", "word3", ...]
}}
Do **not** write any code. Use your linguistic and scientific knowledge to analyze the text.

**User**:

{WMDP-Bio forget set}

**Assistant**:

{response}

---

# F   Additional Details of Keyword Explanation Experiment

In addition to demonstrating the unlearning effectiveness of keyword-based unlearned models in Sec. 4, we further analyze the overlap between keywords extracted from the entire forget set (100%) and those obtained from RANDOM-based coresets. This comparison helps assess whether randomly selected coresets reliably capture the keywords in the full set. For a particular coreset, let the keywords be denoted by $K_c$ and let all the keywords extracted be denoted $K_f$. Then for that coreset, we define the normalized overlap as $\frac{K_c \cap K_f}{|K_f|}$. As seen in **Fig. F1**, the overlapping keywords are higher than the fraction of data present in the coreset. his suggests that even information-agnostic random selection can yield a surprisingly high ratio of keyword overlap with the full forget set. For instance, a 5% RANDOM-based coreset captures approximately 14% of the full set's keywords. We hypothesize that this amplified presence of high-impact keywords is a key factor enabling effective unlearning, even when using such a small, randomly selected subset.

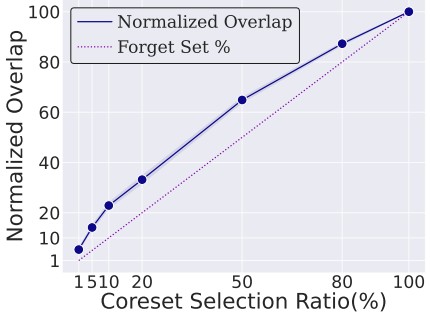

Figure F1: Normalized overlapping ratio (%) of keywords between RANDOM-based coresets and the full forget set for WMDP-Bio. The dotted line represents the percentage of data present in the coreset from the forget set. Results are reported in the form $a \pm 2b$, where $a$ is the mean and $b$ is the standard deviation, computed over 3 independent trials.

# G    Additional Utility Evaluation of Unlearned LLMs Using Coresets.

As shown by the coreset unlearning performance in standard benchmarks (Sec. 4), a surprising finding is that coreset unlearning can achieve lossless UE. This naturally raises the question: Could coreset unlearning offer utility benefits as it negates the influence of fewer forget data points, thereby better preserving the model's performance on general tasks? However, we did not observe a clear UT advantage in existing benchmarks, likely due to the limited scope of their utility evaluations, *e.g.*, MMLU for WMDP and KnowMem on the retain set for MUSE, which may not fully capture the broader utility landscape of the coreset-unlearned models.

Therefore, we conduct additional utility evaluations of coreset-unlearned models, focusing on two tasks inspired by the emergent abilities of LLMs (Wei et al., 2022): (1) a math addition/subtraction task involving 2–5 digit arithmetic that is orthogonal to the unlearning objective, and (2) the TruthfulQA task that assesses factual consistency and truthfulness in LLM responses and may be inadvertently affected by unlearning. In **Fig. G2**, we report the zero-shot accuracy of the aforementioned utility metrics for RANDOM-based coreset-unlearned models using RMU and NPO on WMDP. RMU consistently maintains strong utility across all evaluations, regardless of the coreset selection ratio. In contrast, NPO shows higher variance; however, we observe that for $n$-digit addition and subtraction, performance may benefit from using a 10% coreset in the WMDP-Cyber setting.

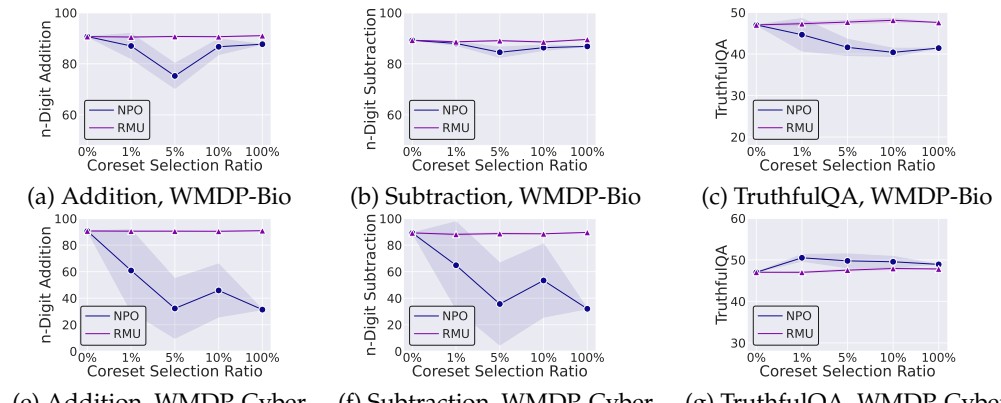

Figure G2: Additional utility evaluation performance for RANDOM-coreset unlearned models using NPO and RMU under Zephyr-7B-$\beta$. (a)-(d) correspond to the performance of a specific utility evaluation (Addition, Subtraction, Truthful QA) of models unlearned using WMDP-Bio or WMDP-Cyber. Here 'Addition' refers to n-digit addition (n=2,3,4,5) and 'Subtraction' refers n-digit subtraction (n=2,3,4,5), where the accuracies are averaged over n. The unlearning task and setting follow the same configuration as in Fig. 2.

