# OpenReview forum: "LLM Unlearning Reveals a Stronger-Than-Expected Coreset Effect in Current Benchmarks"
_colmweb.org/COLM/2025/Conference — COLM 2025_

### Official Review · Reviewer_sYCW · 2025-04-25

**Rating:** 3
**Confidence:** 4
**Ethics Flag:** 1

**Summary:**

The paper demonstrates that for two unlearning benchmarks, using just 5% of the training data (a randomly selected "coreset"), almost the same unlearning rate can be achieved, while better general LM capabilities (measured bh MMLU performance, or the retain set performance).

**Questions To Authors:**

None

**Reasons To Accept:**

Having more data efficient ways to achieve the same outcome is valuable.

The experiments in the paper are carried out competently. Two different datasets are used, with similar results.

The paper explains methods and background very well.

**Reasons To Reject:**

Since no new methods are proposed for using just 5% of the training data, this experimental finding says more about the redundancy in the training data - something that is not discussed at all in the paper. So, when I checked these sets, I found that WMDP handles very specific biochemical weapons content, a very narrow domain, and unlearning for this domain should have little impact on the very different MMLU test set on which preservation of general LM capability is preserved. The examples shown in Table 2, on the other hand, show dramatic degradation of the LM's ability to generate text on the unlearning topic - basically producing word noise, not (what I would have expected) banal and harmless basic facts on the broader topic.

The MUSE data set's training data are the Harry Potter books - again something that is quite redundant when the goal is to not reproduce content from the book. Here is the dataset comes with a retain set of Fandom content - much closer to the content of the books (which is good), but unlearning also has a stronger degradation impact on the retrain set.

Overall, I am left with the impression that the main finding of the paper is that if you have highly redundant training data, you do not need all of it - which is rather obvious, and not a particular interesting finding on the topic of unlearning.

---

> ### Author Response · Authors · 2025-06-01
> **Response to Reviewer sYCW (Part III)**
>
> **Q4: I found that WMDP handles very specific biochemical weapons content, a very narrow domain, and unlearning for this domain should have little impact on the very different MMLU test set on which preservation of general LM capability is preserved.**
>
>
> **A4:** We refer reviewer to A2 for our viewpoint on WMDP.
>
> In addition, we note that unlearning algorithms such as RMU and NPO modify the model weights directly.  Even if one considered WMDP as a “narrow” domain, the removal of knowledge from such a narrow domain does not guarantee that all other knowledge is preserved, even if they are seemingly unrelated in nature (due to the model’s performance sensitivity to weight change). In fact, there exists an inherent trade-off between preserving overall model utility and achieving effective unlearning, as demonstrated in **Figure 10** [7]. The tradeoff between unlearning capability and model utility has been well documented throughout the unlearning literature across datasets and methods, as illustrated in **Figure 5** [9], **Figure 5** [10].
>
> **Q5: The examples shown in Table 2, on the other hand, show dramatic degradation of the LM's ability to generate text on the unlearning topic - basically producing word noise, not (what I would have expected) banal and harmless basic facts on the broader topic.**
>
> **A5:**
> The observation of word-level noise in model outputs when prompted with unlearning-related topics is an expected consequence of the state-of-the-art unlearning techniques such as RMU and NPO. These methods are designed to intentionally degrade the model’s generation capability on queries associated with the unlearning target. For example, RMU achieves this by mapping internal representations corresponding to forget data to random features, effectively erasing semantic alignment. Similarly, NPO forces the model to deviate from its pretrained state using a preference optimization framework based solely on negative forget samples, which suppresses target-related generation. As a result, the model often produces incoherent or noisy outputs when queried about the forgotten content, reflecting the intended forgetting behavior.
>
> Second, it is inherently difficult to ensure that unlearning over the forget dataset allows the model to generate unlearning-irrelevant yet broad, harmless responses when queried on related topics. This difficulty stems from the challenge of precisely defining the *unlearning* scope, that is, removing the influence of in-scope (to-be-forgotten) data while preserving the model’s performance on out-of-scope (retained) content [11]. In practice, delineating the exact boundaries of in-scope examples for generation is nontrivial. For instance, when prompted with a harmful biohazard-related query, it is insufficient for the model to merely avoid the specific target response; it is also critical that the model not produce tangential yet still biohazardous information. This ambiguity complicates what constitutes '*banal and harmless*' content in the context of safety-critical unlearning. We stress that this reflects a broader, well-known challenge in practical machine unlearning and does not represent a flaw in our coreset methodology.
>
> [9] Negative Preference Optimization: From Catastrophic Collapse to Effective Unlearning\
> [10] MUSE: Machine Unlearning Six-Way Evaluation for Language Models\
> [11] Rethinking Machine Unlearning for Large Language Models

---

> > ### Comment · Reviewer_sYCW · 2025-06-02
> > **Locality of unlearning**
> >
> > There are several unlearning papers that not only measure how much targeted responses are suppressed - but also if very related (neighborhood) questions can still be answered correctly. So, this is quite different from producing gibberish on related questions.

---

> > > ### Author Response · Authors · 2025-06-04
> > > **Further Response**
> > >
> > > **Q8: There are several unlearning papers that not only measure how much targeted responses are suppressed - but also if very related (neighborhood) questions can still be answered correctly. So, this is quite different from producing gibberish on related questions.**
> > >
> > > **A8:**
> > > As we clarified in the 4th point of A6, we would like to reiterate that RMU and NPO are currently regarded as state-of-the-art methods for LLM unlearning on established benchmarks such as WMDP and MUSE [1,2,3]. As previously explained, these methods can produce incoherent responses to forget-set queries due to their design: RMU, for example, maps internal representations to random features, while NPO suppresses generation via negative preference optimization. This is how successful unlearning is defined in those approaches, namely, the degeneration of forget-relevant responses without imposing constraints of generating forget-irrelevant responses against unlearning-related input prompts. While alternative approaches such as DPO or IDK-style methods (which return “I don’t know”) may yield more coherent outputs on forget-related prompts, they are fundamentally constrained by their reliance on positive preference data and, as shown in Figure 5 [1], they typically do not match the forgetting effectiveness performance of NPO. \
> > > We would greatly appreciate it if the reviewer could kindly specify which unlearning methods they consider to be stronger or more appropriate for WMDP and MUSE. We would be happy to incorporate such baselines to further validate and extend our findings.
> > >
> > >
> > > [1] Negative Preference Optimization: From Catastrophic Collapse to Effective Unlearning\
> > > [2] Simplicity Prevails: Rethinking Negative Preference Optimization for LLM Unlearning\
> > > [3] The WMDP Benchmark: Measuring and Reducing Malicious Use With Unlearning

---

> ### Author Response · Authors · 2025-06-01
> **Response to Reviewer sYCW (Part II)**
>
> **Q2: Not a particular interesting finding on the topic of unlearning.**
>
>
> **A2:**  We respectfully disagree that this is not an important effect for unlearning. As we mentioned in **Section 6**, there are multiple reasons why this is an **important effect**
> - We start with taking unlearning using the WMDP benchmark as an example, which was developed for unlearning hazardous knowledge in biosecurity, cybersecurity [7]. This is an important task for AI safety, where unlearning has been deemed as a possible defense against AI misuse risks [8]. In this case, a strong coreset effect on such benchmarks indicates that unlearning on current benchmarks may be easier than we expected. This points to **important implications** like the need for richer, more diverse  forget data to better reflect realistic unlearning challenges and the need for more  careful evaluation of the removal of such hazardous knowledge.
> - Our keyword-curated forget set was able to explain a large portion of the unlearning performance. This highlights the need for a deeper investigation of the mechanism of unlearning, which would give us further important intuition about current unlearning algorithms.
> - The existence of the coreset effect in LLM unlearning suggests that unlearning could be pushed into a low-data regime (with computation efficiency benefits for unlearning) if such a coreset exists and can be reliably identified. Remarkably, on current benchmarks, our results show that identifying such coresets is not difficult, as even random selection often suffices.
> The significance of our work has also been recognized by Reviewer xsZY, who remarked:
>     > “The finding that current forget sets are highly redundant has practical significance: it suggests that reliable unlearning could be far cheaper than previously assumed in the unlearning community and could suggest redesigning unlearning benchmarks.”
>
>
> [7] The WMDP Benchmark: Measuring and Reducing Malicious Use With Unlearning\
> [8] An approach to technical agi safety and security
>
>
>
> **Q3: So, when I checked these sets, I found that WMDP handles very specific biochemical weapons content, a very narrow domain; The MUSE data set's training data are the Harry Potter books - again something that is quite redundant when the goal is to not reproduce content from the book.**
>
> **A3:** While WMDP is indeed focused on hazardous biosecurity-related knowledge, we respectfully disagree with the characterization that it represents a narrow domain. On the contrary, WMDP targets one of the most critical and high-impact challenges in LLM safety: preventing the malicious use of publicly available language models. For further context, we refer the reviewer to https://www.wmdp.ai/. This makes WMDP a highly valuable benchmark for assessing real-world risks posed by pre-trained LLMs. Importantly, WMDP differs from other unlearning benchmarks such as MUSE and TOFU in a key aspect: the sensitive knowledge in WMDP is already embedded in the pre-trained model, rather than being enforced via model fine-tuning before unlearning. This distinction makes WMDP a more realistic and challenging testbed for evaluating unlearning methods, as it reflects naturally acquired knowledge rather than synthetic memorization.
>
> Yes, MUSE-Book includes data from the Harry Potter series. However, without a deep and systematic exploration like ours, we hesitate to make a definitive claim that MUSE is inherently redundant when the goal is to not reproduce content from the book. Unlearning, particularly knowledge unlearning, is a fundamentally challenging task. Knowledge can be reflected through diverse yet related facts and narratives within the books, making it difficult to comprehensively erase such information with minimal forget data. In this context, successfully unlearning Harry Potter-related knowledge (as measured by KnowMem in MUSE) **may** require a larger portion of the forget set to sufficiently cover the knowledge scope. This underscores why we believe the coreset effect in MUSE is highly non-trivial: redundancy cannot be inferred purely from the “appearance” of data. Rather, it must be evaluated in light of the specific unlearning objective and its complexity.

---

> > ### Comment · Reviewer_sYCW · 2025-06-03
> >
> > I do not think we have any fundamental agreement on the facts. The disagreement is on the significance of the results, which is a matter of opinion.
> >
> > My view is that the fact that the only way to assess the proper size of the coreset has "evaluated in light of the specific unlearning objective and its complexity" means in practice that you have to unlearn on different sized subsets, which is possibly computationally more expensive than to train on all of it, so it is not clear what is gained here in practice.

---

> > > ### Author Response · Authors · 2025-06-04
> > > **Further Response**
> > >
> > > **Q7: I do not think we have any fundamental agreement on the facts. The disagreement is on the significance of the results, which is a matter of opinion.\
> > >  My view is that the fact that the only way to assess the proper size of the coreset has "evaluated in light of the specific unlearning objective and its complexity" means in practice that you have to unlearn on different sized subsets, which is possibly computationally more expensive than to train on all of it, so it is not clear what is gained here in practice.**
> > >
> > > **A7:**
> > > We are sorry to hear that the reviewer feels there is no fundamental agreement. However, we respectfully encourage a closer examination of our results within the specific context of LLM unlearning, rather than attributing the observed effects solely to assumed data redundancy. As clarified in our earlier responses, the core contribution of our work lies in uncovering and systematically justifying a strong and previously unrecognized coreset effect in LLM unlearning. This phenomenon, as discussed in our analogy to neural network weight pruning (see the 2nd point in A6), highlights a surprising level of data efficiency that challenges prior assumptions and offers new insights into LLM unlearning (see the 4th point in A6).
> > >
> > > Regarding the concern about computational overhead in identifying the “proper” coreset size, we would like to clarify that our approach is not less practical nor more computationally expensive than full forget set unlearning. We do **not** have to unlearn on different sized subsets.
> > > - Random selection, as explored in our study, is computationally minimal, requiring no tuning or search. Practitioners can simply sample a small portion (e.g., 10%) of the full forget set and still achieve strong unlearning performance. Furthermore, our keyword-based analysis (Section 4) supports the feasibility of leveraging lightweight, heuristic approaches, such as using LLMs-as-a-judge, to identify the keyword-guided forget subset.
> > > - For non-random, more sophisticated coreset selection methods (e.g., GraNd, Moderate, Min-K% Prob; see Table 1), the selection process is conducted **once** prior to unlearning. This introduces only a small computational overhead compared to the unlearning step itself (e.g., running RMU or NPO), which dominates the overall cost. Thus, our proposed coreset strategy remains practically viable.
> > >
> > > Lastly, when we previously noted that the coreset must be “evaluated in light of the specific unlearning objective and its complexity,” our intent was not to suggest an exhaustive or expensive coreset search. Rather, we aimed to emphasize that coreset *effectiveness* and *the triviality of its existence* should be interpreted within the context of the unlearning task. Here superficial attributes (e.g., appearance, class labels) *alone are insufficient* to identify coreset effect in LLM unlearning (as supported by the lagging behavior to justify coreset effect during unlearning optimization in Figure 1).

---

> ### Author Response · Authors · 2025-06-01
> **Response to Reviewer sYCW (Part I)**
>
> We thank the reviewer for your valuable feedback. However, we feel that the significance and novelty of our work may have been somewhat overlooked. Please find our detailed responses to your concerns below.
>
>
>
> **Q1:\
> (a) Since no new methods are proposed for using just 5% of the training data, this experimental finding says more about the redundancy in the training data - something that is not discussed at all in the paper.\
> (b) Overall, I am left with the impression that the main finding of the paper is that if you have highly redundant training data, you do not need all of it - which is rather obvious**
>
>
> **A1:** We would like to clarify the confusion around the redundancy of the forget set and emphasize the novelty of our paper.
>
> First, it is important to distinguish between *superficial similarity* (by looking) and *hidden* *redundancy* (revealed through deeper exploration) in training data. Although the proposed coreset identification in the context of unlearning reveals redundancy in the forget data, this belongs to the latter category: As illustrated in **Figure 1**, this coreset effect in unlearning exhibits a lagging behavior, emerging only after extended unlearning training. Identifying such redundancy is highly non-trivial and requires careful investigation and justification (see detailed discussion later). Moreover, literature beyond the unlearning domain also recognizes the challenges of coreset learning. For example, the problem of reducing a training set to a smaller subset without significantly degrading model performance has been extensively studied in computer vision [1,2,3] and more recently for LLMs [4,5,6]. These works consistently highlight that identifying such informative subsets is a non-trivial and a valuable research problem. We refer the reviewer to Section 2 (Coreset Selection) for additional discussion.
>
> Second, we would like to stress that one cannot simply deem data points redundant based on superficial inspection (e.g., simply observing that data points belong to the same class). Our coreset findings are **novel** and **non-trivial**:
> - The fact that unlearning is successful using a subset of the forget set can only be observed if we unlearn for higher than usual epochs than the standard settings, as shown in **Figure 1**. For low selection ratios, the unlearning effectiveness suddenly increases at later epochs. This is a novel finding and has not been observed in previous literature.
> - This coreset effect is surprisingly strong as random selections from the forget set may suffice for unlearning (**Figure 2**, **Table 1**, **Table D1**), which contrasts with standard coreset selection methods that typically require carefully chosen subsets.
> - The rationale behind this strong coreset effect is justified from the perspective of keyword forgetting in current LLM unlearning (**Figure 3**).
> - The quality of coreset unlearning vs. full set unlearning is also justified through mode connectivity, adversarial/relearning robustness, and utility perspectives (Sec. 5).
> These contributions have also been acknowledged by Reviewer xsZY, who remarked:
>     > “The manuscript is clearly written and logically organised. Its exploration of a coreset phenomenon in LLM unlearning is, to the best of my knowledge, novel, and the keyword‑level analysis supplies an intuitive token‑level explanation for the effect.”
>
>
> [1] Deep learning on a data diet: Finding important examples early in training\
> [2] An empirical study of example forgetting during deep neural network learning\
> [3] Moderate coreset: A universal method of data selection for real-world data-efficient deep learning\
> [4]  Data pruning for efficient model pruning in
> neural machine translation\
> [5] STAFF: Speculative coreset selection for task-specific fine-tuning\
> [6] Less: Selecting influential data for targeted instruction tuning

---

> > ### Comment · Reviewer_sYCW · 2025-06-02
> > **Coreset selection**
> >
> > I am not very familiar with the coreset work in computer vision besides being aware that it is a well-studied topic. My main objection is that random selection of a subset is not a very sophisticated method.
> >
> > One may disagree about the importance of the findings of this paper and I leave it to the area chair to assess it.
> >
> > My main objections still stand:
> > [1] That unlearning still works with 5% random samples points to the redundancy of the data
> > [2] The unlearning method is too aggressive, i.e., it does not preserve the neighborhood of the edits.

---

> > > ### Author Response · Authors · 2025-06-04
> > > **Further Response**
> > >
> > > **Q6 : I am not very familiar with the coreset work in computer vision besides being aware that it is a well-studied topic. My main objection is that random selection of a subset is not a very sophisticated method.\
> > > One may disagree about the importance of the findings of this paper and I leave it to the area chair to assess it. \
> > > My main objections still stand: [1] That unlearning still works with 5% random samples points to the redundancy of the data; [2] The unlearning method is too aggressive, i.e., it does not preserve the neighborhood of the edits.**
> > >
> > > **A6 :**  Thank you for the follow-up response.
> > >
> > > First, we would like to clarify that our analysis is not limited to random selection. To validate the robustness of the coreset phenomenon, we had indeed implemented and evaluated multiple more sophisticated selection strategies, including GraNd, Moderate, and Min-K% Prob (see **Table 1** and **Table D1**). As shown, the identified coreset effect consistently holds across these diverse methods, reinforcing that the effect reflects a broader characteristic of current unlearning benchmarks.
> > >
> > >
> > >
> > > Second, our primary goal in this work is not to propose a new or sophisticated coreset selection algorithm, but rather to highlight and systematically validate the surprisingly strong coreset effect in LLM unlearning even when using random data selection. To draw an imperfect but illustrative analogy : this is akin to the early observation in neural network pruning that substantial sparsity can be achieved even through random weight pruning [1,2]. Similarly, our finding reveals that unlearning can succeed with randomly chosen forget data. Once again, as we have clarified in earlier responses, uncovering this coreset effect in LLM unlearning is both non-trivial and impactful.
> > >
> > > Third, regarding the comment that “unlearning still works with 5% random samples points to the redundancy of the data,” we would like to clarify that such redundancy is **not known a priori**, nor has it been established in prior LLM unlearning studies. To the best of our knowledge, no existing work has systematically examined how much forget data is actually required to achieve high-quality unlearning under standard benchmarks. Our study is the first to reveal that a strong coreset effect emerges only after extended unlearning training, exhibiting a lagging effect relative to full forget-set unlearning (as shown in Figure 1). We also provide an explanation for this phenomenon through a keyword-level analysis (Section 4) and validate the quality of coreset unlearning through multiple dimensions. We believe these findings raise important future research directions about unlearning in low-data regimes and call for revisiting the design and evaluation rigor of existing unlearning benchmarks.
> > >
> > > Fourth, regarding the comment that “the unlearning method is too aggressive, i.e., it does not preserve the neighborhood of the edits,” we respectfully disagree with the implication that this undermines the value of our findings. The unlearning methods we used, NPO and RMU, are widely regarded as state-of-the-art in the literature, as they have demonstrated superior performance on standard evaluation metrics and are officially benchmarked in both WMDP and MUSE. While we acknowledge that the current successful unlearning using these methods may not fully preserve semantic neighborhoods in generation, they nonetheless represent the current best practices in the field and serve as the standard reference point for evaluating unlearning efficacy.
> > >
> > >
> > > Furthermore, if the reviewer is aware of alternative unlearning approaches that are considered state-of-the-art for the WMDP or MUSE benchmarks, we would be glad to include these methods and assess whether the coreset effect similarly holds for them.
> > >
> > > [1] Han, Song, Huizi Mao, and William J. Dally. "Deep compression: Compressing deep neural networks with pruning, trained quantization and huffman coding." arXiv preprint arXiv:1510.00149 (2015).
> > >
> > > [2] Chijiwa, Daiki, et al. "Pruning randomly initialized neural networks with iterative randomization." Advances in neural information processing systems 34 (2021): 4503-4513.

---

### Official Review · Reviewer_wji5 · 2025-05-11

**Rating:** 6
**Confidence:** 5
**Ethics Flag:** 1

**Summary:**

The paper demonstrates that unlearning outcomes achieved with the full forget set can often be preserved using a randomly selected subset as small as 5%, indicating that current LLM unlearning tasks may be surprisingly easy in low-data regimes The paper attributes this coreset effect to a keyword-driven mechanism, where a small number of high-impact tokens dominate unlearning efficacy. The authors further validate the faithfulness of coreset-unlearned models using mode connectivity analysis and robustness to jailbreaking attacks.

**Questions To Authors:**

Could you evaluate your method on the ToFU benchmark, following the experimental setup used in SOUL? This would provide a more comprehensive assessment of SoTA method’s unlearning effectiveness.

**Reasons To Accept:**

1. The authors identify and analyze the coreset effect in LLM unlearning and reveal a fundamental property of two benchmarks. The coreset effect demonstrates that effective unlearning can occur with extremely small subsets, highlighting opportunities for computational efficiency.

2. The authors provide an explanation via token-level analysis, attributing unlearning success to high-impact keywords, which advances understanding of model behavior.

**Reasons To Reject:**

The paper does not include experiments on the ToFU benchmark following SOUL [1] to evalute Forgetquality, Acc., Rouge-L, and MIA, which is essential for evaluating unlearning performance in language models. Including results on SOUL would strengthen the empirical validation and allow for more direct comparisons with recent unlearning methods.

[1] Jia, Jinghan, et al. "SOUL: Unlocking the Power of Second-Order Optimization for LLM Unlearning." Proceedings of the 2024 Conference on Empirical Methods in Natural Language Processing. 2024.

---

> ### Author Response · Authors · 2025-06-01
> **Response to Reviewer wji5**
>
> We sincerely thank you for your constructive feedback and appreciate your positive remarks. Below, we provide detailed responses to address your concerns point by point.
>
>
> **Q1: The paper does not include experiments on the ToFU benchmark following SOUL [1] to evalute Forgetquality, Acc., Rouge-L, and MIA, which is essential for evaluating unlearning performance in language models.**
>
> **A1:** We agree that incorporating results using SOUL would strengthen our findings on the coreset effect, and we thank you for this valuable suggestion. In response, we have conducted the additional experiments and will include the results in our response A2.
>
>
>
> However, we chose MUSE as one of our primary unlearning benchmarks over TOFU for the following reasons.
> - From our perspective, MUSE and TOFU serve similar unlearning purposes: both are designed to support data unlearning (although MUSE also covers evaluation of knowledge unlearning). This stands in contrast to WMDP, which focuses on knowledge unlearning. We refer the reviewer to Section 3, line 182 for a detailed discussion on these two types of unlearning.
> - MUSE is constructed from a large-scale, real-world corpus comprising news articles and books, offering a more realistic setting for evaluating unlearning. In contrast, TOFU is based on synthetic autobiographical data, which may not adequately capture the complexities of practical unlearning scenarios.
> - MUSE used privacy leakage (PrivLeak) metric to assess the extent to which an unlearned model exposes membership information; specifically, whether it reveals that the forget set was part of the original training data. In contrast, TOFU introduces the forget quality metric, defined as a p-value that quantifies the similarity between an unlearned model and a retrained model (trained from scratch without the forget data). A small p-value indicates significant dissimilarity, suggesting that the current unlearning method fails to approximate retraining, the gold standard for data removal from a privacy perspective. However, when the p-value exceeds 0.1, the differences between different unlearning methods become statistically insignificant, making it difficult to draw reliable comparisons or establish a meaningful ranking.
>
>
> **Q2: Including results on SOUL would strengthen the empirical validation and allow for more direct comparisons with recent unlearning methods.**
>
> **A2:**  [Table R1](https://ibb.co/hnt8Q3b) presents the coreset unlearning performance for MUSE using SOUL-based NPO (SOUL-NPO) across the RANDOM-based coreset selection method. We use the standard hyperparameter settings for the second-order optimizer used in [1] and set a learning rate of 5e-06 for all our experiments. Consistent with the findings in our paper, RANDOM-based coreset selection performs comparably to full forget set unlearning. In some cases, it even surpasses it. For instance, on MUSE-Books (Table R1, column 5), using just 10% of the forget data yields significantly lower PrivLeak compared to unlearning with the full forget set. In addition, compared to the first-order optimization-based NPO in our submission (Table D1), SOUL-based NPO demonstrates slightly better performance, as reflected by higher utility and lower PrivLeak in full forget set unlearning, consistent with the findings reported in [1].
>
>
> [1] SOUL: Unlocking the Power of Second-Order Optimization for LLM Unlearning

---

> > ### Comment · Reviewer_wji5 · 2025-06-09
> >
> > The rebuttal addressed my concerns. Hence, I keep my original score.

---

> > > ### Author Response · Authors · 2025-06-10
> > > **Thank you**
> > >
> > > Dear Reviewer wji5,
> > >
> > > We are glad to hear that our rebuttal has addressed your concerns. We sincerely appreciate your thoughtful feedback and valuable comments. Thank you again for recognizing the contributions of our work.
> > >
> > > Authors

---

### Official Review · Reviewer_xsZY · 2025-05-12

**Rating:** 7
**Confidence:** 3
**Ethics Flag:** 1

**Summary:**

The paper probes whether unlearning a large language model truly requires the full forget set or whether a tiny “coreset” (≤ 5 %) can achieve comparable erasure. Across two hazardous‑knowledge and two memorization benchmarks, three 7‑billion‑parameter models, and two representative algorithms, the authors show that coreset‑based unlearning preserves utility while matching—or occasionally exceeding—full‑set performance on erasure metrics.

The manuscript is clearly written and logically organised. Its exploration of a coreset phenomenon in LLM unlearning is, to the best of my knowledge, novel, and the keyword‑level analysis supplies an intuitive token‑level explanation for the effect. The finding that current forget sets are highly redundant has practical significance: it suggests that reliable unlearning could be far cheaper than previously assumed in the unlearning community and could suggest redesigning unlearning benchmarks.

**Reasons To Accept:**

- Strong empirical evidence. The coreset effect is validated across two datasets, three models, two representative unlearning methods, and four coreset‑selection strategies, robustly supporting their main claim.
- Novel and impactful result. Showing that only 5 % of the forget set can achieve comparable unlearning performance is both surprising and new, and is likely to have significant impact on the LLM‑unlearning community.

**Reasons To Reject:**

- In Section 5, the linear‑mode‑connectivity analysis suggests that the optimization outcomes of the models are nearly identical, yet Table 3 shows divergent results for the downstream fine‑tuning‑based knowledge‑recovery attack. The paper does not explain this inconsistency.

---

> ### Author Response · Authors · 2025-06-01
> **Response to Reviewer xsZY**
>
> We sincerely thank you for your careful review and greatly appreciate your positive remarks about our contribution. We address your concern below:
>
> **Q1: In Section 5, the linear‑mode‑connectivity analysis suggests that the optimization outcomes of the models are nearly identical, yet Table 3 shows divergent results for the downstream fine‑tuning‑based knowledge‑recovery attack. The paper does not explain this inconsistency**
>
> **A1:**
>
> First, we would like to clarify that Table 3 reports robustness to input-level jailbreaking attacks, whereas robustness to downstream fine-tuning is presented in Figure 5.
>
> Second, although Figure 4 suggests that the models are linearly mode-connected, indicating convergence to similar optima, they are not identical. The visual similarity arises from the coarse y-axis scale (0–100%), which masks subtle differences. We will revise Figure 4 to adopt a finer resolution and constrain the y-axis range to 60–80% to better reflect the variations.
>
> Third, as noted above, the mode-connected models do *not* share *exactly* the same initial model state prior to fine-tuning across different coreset ratios. As a result, e.g., in **Figure 5a**, downstream fine-tuning on GSM8K can yield different optimization trajectories for different coreset ratios. To support this, we refer the reviewer to [1], which shows that even when models are fine-tuned on the *same dataset* from the *same pretrained model*, they can converge to different loss basins and yield distinct generalization performance. This variability arises not only from model initialization or dataset, but also from differences in the fine-tuning optimization dynamics (e.g., data sampling), which *cannot* lead to the same loss values and performance outcomes.
>
>
> [1] Linear Connectivity Reveals Generalization Strategies

---

> > ### Comment · Reviewer_xsZY · 2025-06-10
> >
> > Dear Authors,
> >
> > I appreciate the author's feedback on my review comments. I hope the authors clarify these points in the final version. I will maintain my positive score.

---

> ### Author Response · Authors · 2025-06-10
> **Thank you and will surely include the clarifications in the revision**
>
> Dear Reviewer xsZY,
>
> Thank you very much for acknowledging our rebuttal effort, and for your thoughtful comments and the detailed summary of our contributions in your review. We will ensure that the clarifications discussed during the rebuttal phase are incorporated into the revised version of the manuscript.
>
> Authors

---

### Decision · Program_Chairs · 2025-07-08

**Decision:**

Accept

**Comment:**

The paper introduces and systematically validates an unexpected "coreset effect" in current LLM-unlearning benchmarks: across unlearning benchmarks, two models and several selection strategies, unlearning with as little as 5 % of the forget set consistently matches, or even surpasses, full-set performance while preserving utility in unlearning, and the authors offer a convincing token-level explanation via high-impact keywords that are sufficiently represented in the core set, plus robustness analyses. Two reviewers deem the work novel and impactful, raising only clarifications on a fine-tuning inconsistency and a missing evaluation. the authors’ rebuttal supplied new results that seem to satisfy those concerns. Reviewer sYCW argues the findings merely expose obvious data redundancy and criticizes the aggressiveness of NPO/RMU. In their response, the authors argue that the effect does not stem from mere surface-form duplication, and point out to the plot that shows rapid emergence of unlearning effectiveness after relatively many epochs. I think the reviewer is correct in pointing out that this seems to be mainly lexical phenomenon, but I am convinced from the authors' argument that it's unknown and nontrivial, and that it does not seem to be mere duplication in the training data.  I think a major contribution here is the suggestion that unlearning is done rather "shallowly" based on a set of topic-related keywords. It would be interesting to connect this to the abrupt but slow emergence in Figure 1.

I think the authors should try to make these points more salient in the text. Weighing the reviews and the discussion, I recommend acceptance: the paper provides new, nontrivial results that can spark additional research on unlearning.